# Sequential Approach for a Critical-View COlectomy (SACCO): A Laparoscopic Technique to Reduce Operative Time and Complications in IBD Acute Severe Colitis

**DOI:** 10.3390/jcm9103382

**Published:** 2020-10-21

**Authors:** Gianluca Matteo Sampietro, Francesco Colombo, Fabio Corsi

**Affiliations:** 1Division of Surgery, ASST Rhodense, Rho Memorial Hospital, 20017 Milano, Italy; gianluca.sampietro@unimi.it; 2Division of Surgery, ASST Fatebenefratelli-Sacco, Luigi Sacco University Hospital, 20157 Milano, Italy; colombo.francesco@asst-fbf-sacco.it; 3Department of Biomedical and Clinical Sciences, University of Milano, 20157 Milano, Italy; 4Breast Unit, Surgery Department, Istituti Clinici Scientifici Maugeri IRCCS, 27100 Pavia, Italy

**Keywords:** inflammatory bowel disease, acute severe colitis, surgery, laparoscopy, surgical technique, colectomy, outcomes

## Abstract

Acute severe colitis is the major indication for surgery in inflammatory bowel diseases (IBD), and in particular, in ulcerative colitis (UC). A laparoscopic approach for abdominal colectomy is recommended, due to better perioperative and long-term outcomes. However, costs, time-spending, and outcomes are still a topic of improvement. We designed a standardized 10-steps, sequential approach to laparoscopic colectomy, based on the philosophy of the “critical view of safety”, with the aim to improve perioperative outcomes (operative duration, estimated blood loss, complications, readmissions, reoperations, and length of postoperative stay). We performed a retrospective cohort study using data from a prospectively maintained clinical database. We included all the consecutive, unselected patients undergoing laparoscopic subtotal colectomy (SCo) for IBD between 2008 and 2019 in a tertiary IBD Italian Centre. Starting from 2015, we regularly adopted the novel Sequential Approach for a Critical-View Colectomy (SACCo) technique. We included 59 (40.6%) patients treated with different laparoscopic approaches, and 86 patients (59.4%) operated on by the SACCo procedure. The mean operating time was significantly shorter for the SACCo group (144 vs. 224 min; *p* < 0.0001). The SACCo technique presented a trend to fewer major complications (6.8% vs. 8.3%), less readmissions (2.3% vs. 13.5%; *p* = 0.01), and shorter postoperative hospital stay (7.2 vs. 8.8 days; *p* = 0.003). Laparoscopic SACCo-technique is a safe and reproducible surgical approach for acute severe colitis and may improve the outcomes of this demanding procedure.

## 1. Introduction

Subtotal colectomy (SCo), in an urgent or emergency setting, is the treatment of choice for severe acute colitis in Inflammatory Bowel Disease (IBD) patients after failure of medical therapy [1,2,3,4]. The laparoscopic approach has been proposed based on several short- and long-term advantages, such as less perioperative complications, short recovery and hospital discharge, fewer adhesions and incisional hernias, enhanced body image, and preservation of fecundity in female patients. Toxic megacolon, massive hemorrhage, and hemodynamic instability are still considered the main contraindications to laparoscopy [1,2,5,6,7,8,9,10,11,12]. If the elevated cost of laparoscopic devices should be compensated by shorter postoperative hospitalization, a longer operating time could be considered a drawback of laparoscopy, compared to open surgery, especially in referral hospitals with a busy urgent list. In fact, a laparoscopic colorectal resection requires substantial experience and advanced laparoscopic skills in patients with IBD, even more, when dealing with acute settings. Furthermore, a prolonged intestinal mucosal ischemia, through the mechanism of systemic toxicity, may increase the risk of complications in acute patients with elevated inflammatory indices, risk of thromboembolism, or under maximal rescue treatment with steroids and biologicals [13,14,15,16,17]. Several technical approaches are currently used by IBD surgeons to perform urgent SCo: Open surgery, multiport versus single-port versus hand-assisted laparoscopic surgery, central vascular ligations versus pericolic dissection, medial-to-lateral versus lateral-to-medial mobilization of the mesentery, rectal stump closure versus rectosigmoid mucocutaneous fistula [11,18,19,20,21,22,23,24,25,26,27,28].

Taking inspiration from “the critical view of safety” approach described for laparoscopic cholecystectomy [29], we designed and systematically applied a ten(X)-steps standardized technique for laparoscopic SCo named Sequential Approach for a Critical-View Colectomy (SACCo).

The aim of the present study was to verify the safety and efficacy of the SACCo technique in reducing surgical operating time and postoperative complications in IBD acute severe colitis.

## 2. Materials and Methods

### 2.1. Study Population

“Luigi Sacco” University Hospital is a tertiary care academic center with a dedicated IBD Surgical Unit as part of a large multidisciplinary team, with more than 8000 IBD patients in follow up. A prospective database for IBD surgery has been established since 1993, and recently reviewed and renamed Ulcerative Colitis and Crohn’s Disease-Clinical Auditing and Research Database (UC- and CD-CARD), with 120 variables, including demographic, history of IBD, indications for surgery, medical therapy, nutritional status, preoperative, perioperative, and long-term follow-up data [30,31,32,33].

All the consecutive, unselected patients submitted to SCo in urgent or emergency settings, from January 2008 to December 2019, were considered eligible for the study. Starting in January 2015, the SACCo technique was systematically adopted, and these patients constitute Group A. All the patients treated from January 2008 to December 2014 with any laparoscopic approach, depending on the operating surgeon attitude and preference, were used as the control group (Group B). The decision for surgical treatment was provided after a formal multi-disciplinary evaluation, involving gastroenterologists, surgeons, pathologists, and radiologists.

### 2.2. Sequential Approach for a Critical-View Colectomy (SACCo) Technique

The patient is positioned in the classical Lloyd-Davies position. Three 5–12 mm trocars are placed in the umbilical region, in the suprapubic region, and in the left iliac fossa. One 5–15 mm trocar is placed in the right iliac fossa, where the ileostomy will be fashioned. One 5 mm trocar is placed in the xiphoid region. The surgeons stand on the patient’s right side.

### 2.3. Phase 1. Left Colectomy

The patient is moved in reverse Trendelenburg position, with the left side tilted up. The gastro-colic ligament is divided starting from the left side of the Henle’s trunk, and the lesser sac is completely opened. Critical-view I: The superior aspect of the pancreas, the splenic hilum, and the sustentaculum lienis. The gastrocolic ligament section and the mobilization of the colon with the omentum allow a fast and easy exposure of the pancreas, the splenic hilum, and the splenic flexure, minimizing the risk of splenic injury. The detachment of the greater omentum from the colon is a time-spending procedure, without any practical advantage, and the preservation of the omentum increases the risk of adhesions at the time of completion proctectomy and ileo-anal-pouch construction. The pancreatico-colic, spleno-colic, and the phrenico-colic ligaments are divided, and the splenic flexure fully mobilized between Toldt’s and Gerota’s fascia. Critical-view II: Inferior margin of the pancreas, intact Gerota’s fascia, pancreas tail, and spleen. A small gauze is positioned in the field, the patient translated in Trendelenburg position, and the splenic flexure is repositioned in its original place (left handkerchief maneuver). The greater omentum is raised above the transverse colon, and the colon retracted superiorly. From medial to lateral, the inferior mesenteric vein is isolated until the dissection reaches the gauze. This maneuver ensures rapid isolation of the vessel, as the inferior margin of the pancreas has already been isolated. Critical-view III: Inferior mesenteric vein. The left handkerchief maneuver is a “hybrid” solution, since it combines the abatement of the splenic flexure, with a superior and lateral approach, and the mobilization of the mesocolon, with a medial-to-lateral approach. This trick reduces the risk of dissecting a wrong plane under Gerota’s fascia laterally to the aorta or under the body and tail of the pancreas, preserving retroperitoneal structures. In obese patients, or in the presence of a thickened mesentery with enlarged lymph nodes, a significant time saving is assured. After the division of the inferior mesenteric vein, the mobilization of the left mesocolon is performed, from the sacrum promontory to the divided inferior mesenteric vein. This rapidly identifies the inferior mesenteric artery, originating directly from the aorta. Critical-view IV: Inferior mesenteric artery. As the mesentery of the left colon pertaining to the inferior mesenteric vein cephalad of the artery is mobilized, the control of the inferior mesenteric artery is safe and easy, and the level of the section, with or without extended lymph nodes dissection, can be decided. This approach also visualizes and preserves of the superior hypogastric plexus. After the division of the artery, a complete mobilization of the left mesocolon, from medial to lateral, up to the left paracolic gutter, is performed. The gutter is left intact, while the colon is being raised, and the retroperitoneum of the left abdominal quadrant visualized. Critical-view V: Left gonadic vessels and ureter under intact Gerota’s fascia. The left paracolic gutter is divided, and the entire left colon mobilized. The inferior mesenteric artery is retracted cephalad, and following its inferior margin, the control and section of the superior rectal artery are easily achieved. It’s of paramount importance to not continue with the dissection beyond the promontory of the sacrum. In fact, a pelvic dissection in acute IBD patients is strongly discouraged, and leaving an adequate rectal stump, with the intact sacral plane, is the basis for a safe completion proctectomy and ileo-anal-pouch anastomosis (as reported by European and Italian surgical Guidelines) [1,2,3,4]. The colon is finally divided at the recto-sigmoid junction by an endo-stapling device. Surgical steps from I to IV are illustrated in detail in Figure 1.

### 2.4. Phase 2. Right and Transverse Colectomy

The surgeons move on to the patient’s left side. The patient is moved in reverse Trendelenburg position, with the right side tilted up. The gastro-colic ligament is divided starting from the right side of the Henle’s trunk, and the superior aspect of the middle colic vessels visualized. Critical-view VI: Henle’s trunk and middle colic vessels. The cholecysto-duodeno-colic and the hepato-colic ligaments are divided, and the hepatic flexure fully mobilized between Toldt’s and Gerota’s fascia. Critical-view VII: Duodenum, intact Gerota’s fascia, gallbladder, and Morrison’s pouch. A small gauze is positioned in the field, the patient translated in Trendelenburg position, and the hepatic flexure repositioned in its original place (right handkerchief maneuver). The terminal ileum is retracted cephalad, peritoneum in incise from the aortoiliac carrefour to the caecum, and the terminal ileum and ascending colon mobilized, from medial to lateral, between Toldt’s and Gerota’s fascia. Mobilization is easily performed until the gauze is visualized. Critical-view VIII: Right gonadic vessels and ureter under intact Gerota’s fascia, inferior margin of the duodenum, pancreato-duodenal region. As for the left colectomy, the right handkerchief maneuver allows a rapid dissection of the right mesocolon without the risk of retroperitoneal structures injury. Once the inferior margin of the duodenum and the duodenal-pancreatic region is reached, the control of the ileo-colic and right colic vessels, the level of their section, and the extension of lymph nodes dissection can be decided. The greater omentum is raised above the transverse colon, and the colon retracted superiorly. With traction of the ileocecal region, the ileocolic pedicle can be seen bowstringing through the mesentery. If no suspicion of colorectal cancer is present, the ileocolic vessels should be preserved, in accordance with European and Italian surgical guidelines, in case of necessity of mesenteric lengthening at the time of ileo-pouch-anal anastomosis [1,2,3,4]. Critical-view IX: Ileocolic vessels, the superior margin in case of preservation, inferior and superior margins for control and division. The right paracolic gutter is divided, and the entire ascending colon mobilized. The terminal ileum is divided by an endo-stapling device close to the ileocecal valve. The patient is translated in reverse Trendelenburg position, the transverse colon is retracted caudally, and the middle colic vessels are clearly visualized in the remaining attachments of the transverse mesocolon. Critical-view X: Middle colic artery and vein. At this point of the SCo, the whole colon is still vascularized from the middle colic vessels, so that no mucosal ischemia, nor ischemic toxicity has happened. The middle colic pedicle should be dissected and divided, or easily separated using an endo-stapling device with the vascular cartridge. This approach to the middle colic vessels, from the left to the right side, after complete dissection of the ascending and descending mesocolon, is not appropriate in case of colorectal cancer, since an adequate lymph nodes dissection and a proximal ligation at the superior mesenteric hilum is not always achievable. However, the dissection of the middle colic vessels along the margin of the superior mesenteric artery and vein is a time-spending procedure, associated with a high risk of vessel injury in case of thickened and retracted mesentery, full of enlarged lymph nodes.

Extraction of the specimen and ileostomy construction are made in the place of the right iliac fossa 5–15 mm trocar. Drainage is positioned in the pelvis from the left iliac fossa 5–12 mm trocar. Surgical steps from VI to X are illustrated in detail in Figure 2.

### 2.5. Outcome Measures

The primary endpoint was to compare the perioperative outcomes and morbidity rates between the two groups. Intraoperative variables included duration of surgery, estimated blood loss, and conversion to open surgery. Postoperative complications were classified according to Clavien-Dindo classification [34], in terms of maximum complication for patients developed during their hospital stay and within 90 days from surgery. Patients receiving total parenteral nutrition (TPN) or blood transfusion in the contest of a scheduled perioperative optimization were not considered as Clavien-Dindo grade II complications in the final analysis [11]. Severe postoperative morbidity was defined as any complication graded III or IV.

### 2.6. Statistical Analysis

Quantitative data were reported as mean and standard deviation. Continuous variables were analyzed using a two-tailed, unpair, Student’s t-test, and proportions were compared using two-tailed, Fisher’s exact or Chi-square test, where appropriate. All tests were two-sided with a level of significance set at *p* < 0.05. Statistical analysis was performed using STATISTICA 8 (data analysis software system), Stat Soft Inc.

The UC- and CD-CARD and the study were approved and conducted according to the ethical standards of the ethics committee of our Institution (approval number 1247703013039), and results are reported according to the Strengthening the Reporting of Observational Studies in Epidemiology (STROBE) guidelines [35].

## 3. Results

From January 2008 to December 2019, a total of 348 patients underwent abdominal colectomy (with a definitive ileostomy, restorative proctocolectomy, or ileo-rectal anastomosis) for IBD at “Luigi Sacco” University Hospital (305 UC, 43 CD). SCo was performed in urgent or emergency settings, for severe acute colitis, in 169 consecutive patients. Twenty-four (14%) patients, operated on with an open approach (due to previous major abdominal surgery or for hemodynamic instability), were excluded from the study. Among the remaining 145 patients, the SACCo technique was adopted in all 86 patients operated on after January 2015 (59.4%, Group A). The 59 patients (40.6%) operated on before this date were treated with an unstandardized laparoscopic approach and made up the control (Group B).

Table 1 summarizes the patient characteristics for the two groups, that were similar in terms of age, gender, smoking habit, IBD family history, and extraintestinal manifestations. There was a statistically significant difference in the use of preoperative anti-TNF therapy between the groups: Patients operated with the SACCo technique received much more frequently this kind of therapy in comparison with patients of Group B (77.9% vs. 28.8%, *p* < 0.0001). Preoperative blood tests (hemoglobin, white blood cell count, C-reactive protein, albumin, and prealbumin) were not statistically different between the groups. TPN before and after surgery was equally distributed (74.4% for SACCo and 71.1% for Group B).

The comparisons of surgical results are reported in Table 2. Duration of surgery was significantly shorter for Group A undergoing the SACCo technique (144.65 ± 37.4 vs. 224,65 ± 52.5 min; *p*< 0.0001). The overall conversion rate was 2%, 1.1% for Group A, and 3.3% for Group B (ns). Postoperative complications were similar between the two groups, with a cumulative serious complication rate (Clavien-Dindo III) of 7.5%, and mortality was nil. While readmission and total hospital staying were similar between the two groups, a significantly shorter postoperative hospitalization and lower readmission were in favor of the SACCo technique.

## 4. Discussion

At “Luigi Sacco” University Hospital, while colorectal resections for cancer and diverticulitis were routinely performed by laparoscopy since 2005, the use of this operative technique was adopted with a significant delay in IBD patients, mainly due to a cultural reticence. From a surgical point of view, IBD is generally considered tricky patients, with impaired anatomy, thickened and easily bleeding mesentery, massive lymphopathy, dilated bowel loops, and general fragile condition, due to heavy pharmacological treatments. In 2008, a definite protocol was adopted to offer a totally laparoscopic approach to the patients needing SCo and restorative proctocolectomy, and the need for a surgical technique designed to minimize operative time and complications became evident, especially for urgent procedures. In fact, the surgical strategy, the use of different energy devices, and the number and site of operative ports varied according to the patient’s characteristics and surgeon preferences. Furthermore, since the dedicated IBD surgeons involved in all laparoscopic procedures were always assisted by trainees and residents, a standardization of the technique also became necessary for teaching purposes.

Emergency Sco for IBD acute colitis is, nowadays, a life-threatening procedure, with reported mortality in Europe and the United States that ranges from 1% to 13%. In the 2012 report from the Danish population-based nationwide cohort study on 2889 patients, mortality after urgent Sco was 5.2% in UC and 8.1% in CD. Expected postoperative morbidity is equally high, ranging from 16% to 65%. However, high volume centers (at least >10 cases per year), with dedicated IBD surgeons, multidisciplinary teams, and aggressive surgical policy, have reported a mortality rate after laparoscopic Sco of <2% [5,26,27,36,37,38,39,40,41,42,43,44,45].

We reported here the feasibility and safety of a novel laparoscopic approach for Sco in IBD patients presenting with severe acute colitis. In daily clinical practice, patients with acute severe colitis complicating an IBD, are frequently malnourished, anemic, and under high doses of steroids or biologicals. In this series, most patients were under multiple medications at the time of their operation, with a median of 2.5 drugs for the patient. The SACCo group presented a significantly more frequent preoperative use of anti-TNF therapy. This aspect is probably due to the close multidisciplinary monitoring of the acute patients, with more confidence from the gastroenterologist in the use of aggressive therapies by having a dedicated surgeon available for an emergency Sco in case of lack of response or worsening of the patient’s conditions. Postoperative mortality of this series was nil, confirming the results of several referral centers, and overall morbidity was 13.7%, comparing favorably with other studies in which morbidity rate ranged between 16% and 65%. Patients treated with the SACCo technique presented a decreased risk for major complications (Clavien-Dindo ≥3) in comparison with the traditional technique, but without reaching a statistical significance (6.8% vs. 8.3%). Our proposed technique appears to be significantly time-saving, probably due to its reproducibility in all clinical settings. The median operative duration for the SACCo technique is shorter than many of the operating times reported elsewhere in the literature and comparable with the surgical time of an open approach [27,36,37,46]. Furthermore, an additional benefit has emerged from a shorter postoperative hospitalization (*p* = 0.003) and lower readmission (*p* = 0.01) compared to all the other laparoscopic approaches. In accordance with the study from Vlug et al., one possible reason for the better results of the SACCo technique should be related to the preservation of the intestinal barrier during surgery, reducing the effect of systemic toxicity [14]. Another aspect should be the reduced risk of thromboembolism, due to central vascular ligation, and the maintenance of the colon vascularity until the end of the procedure when the middle colic vessels are ligated (Critical-view, step X) [15,16].

In comparison to most reported series in UC surgery, we had a very high laparoscopic rate (86% of SCo from 2008) [11,39,47,48]; while usually more than half of SCo are still performed by an open approach, as recently confirmed by large databases [49,50]. On the contrary, the conversion rate was quite low (1.1%), especially considering that reported conversion rate in trials and observational studies ranges from 1 to 23% [8,24,39,45,51,52,53,54], and that conversion rate after laparoscopic cholecystectomy, the most popular laparoscopic procedure, is reported to be 5–10% [55].

Learning from our previous experiences and applying the “critical view of safety” philosophy, we simplified this demanding procedure to obtain a time-saving, reproducible, and safe, laparoscopic approach to perform SCo—“the SACCo procedure”. Laparoscopic colectomy procedure is still a very little standardized operation, with several different variations and modifications depending on the surgeon’s preference or experience. This rationalization of the surgical technique aims to avoid mistakes, due to anatomical alterations and altered visual perception, thus favoring an easy and reproducible execution of the main surgical steps. The benefit of a standardized technique as the SACCo procedure is especially evident when risk factors are present, such as acute inflammatory conditions, pathological mesentery, or anatomical variations.

This series has several limitations: It is a single, tertiary center experience with a limited number of cases. The two groups were distributed in two different time periods (before and after 2015). However, from a technical point of view, the SACCo technique showed some relevant tips, with a reliable impact on the clinical management of patients.

## 5. Conclusions

The Sequential Approach for a Critical-View Colectomy (SACCo) technique is a safe and effective procedure for acute severe colitis in IBD patients. A complex and demanding procedure, as the SCo, has been standardized with a significant reduction in operating time, and a clinical advantage in terms of postoperative outcome.

## Figures and Tables

**Figure 1 jcm-09-03382-f001:**
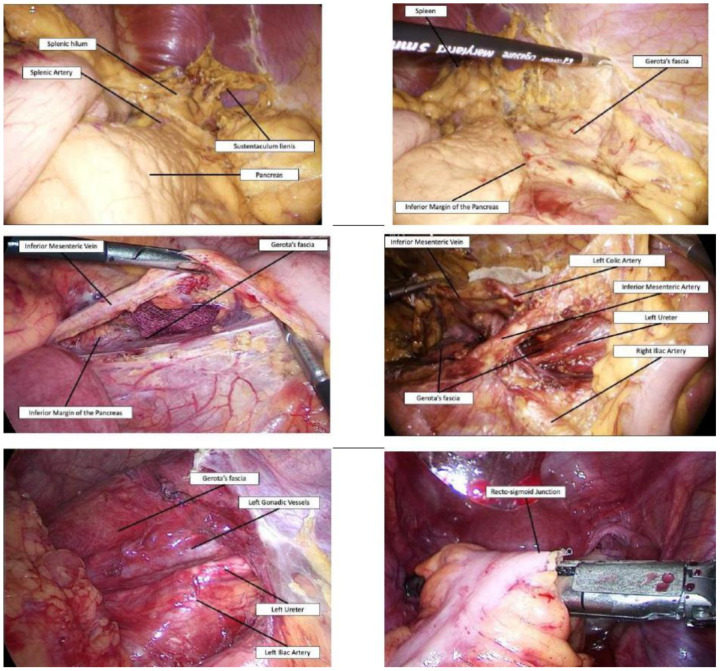
Upper panel: Critical-view I (left) and II (right). Middle panel: Critical-view II (left) and III (right). Lower panel: Critical-view IV (left) and rectosigmoid division (right).

**Figure 2 jcm-09-03382-f002:**
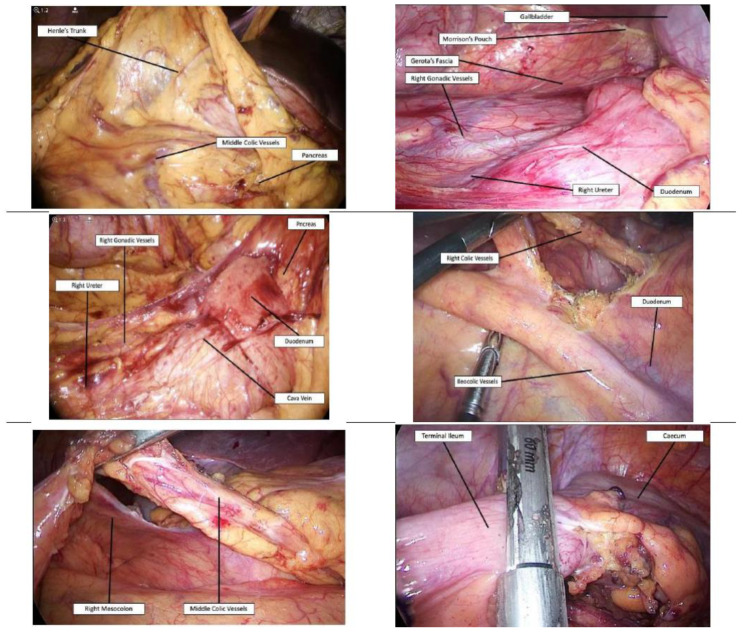
Upper panel: Critical-view VI (left) and VII (right). Middle panel: Critical-view VIII (left) and IX (right). Lower panel: Critical-view X (left) and terminal ileum division (right).

**Table 1 jcm-09-03382-t001:** Patients characteristics.

	Group A (*n* = 86)	Group B (*n* = 59)	*p*
Age (years)	42.8 ± 17.3	45 ± 16.3	ns
Gender			
M	47 (54.6%)	37 (62.7%)	
F	39 (45.4%)	22 (37.3%)	ns
Age at Diagnosis (years)	35.2 ± 18.06	38.3 ± 17.5	ns
Disease Duration (years)	7.19 ± 7.3	7 ± 7.2	ns
Smoking Habit	2 (2.3%)	5 (8.4%)	ns
Family History	6 (6.9%)	3 (5%)	ns
Extraintestinal Manifestations	11 (12.7%)	7 (11.8%)	ns
Hb (g/L)	10.6	10.4	ns
WBC count (×10^9^/L)	10.543	8.971	ns
CRP (mg/L)	42.82	37.4	ns
Albumin (g/L)	3.7	2.8	ns
Transthyretine (g/L)	0.17	0.14	ns
TPN	64 (74.41%)	42 (71.1%)	ns
Preoperative Therapy			
5-ASA	6 (6.9%)	5 (8.4%)	ns
Immunomodulators	18 (20.9%)	10 (16.9%)	ns
Biologicals	67 (77.9%)	17 (28.8%)	<0.0001
Steroids	74 (86%)	49 (83%)	ns
Combo	29 (33.7%)	17 (28.8%)	ns

ns, not significant; Hb, haemoglobin; WBC, white blood cell; CRP, C-reactive protein; TPN, total parenteral nutrition; 5-ASA, 5-aminosalicylic acid.

**Table 2 jcm-09-03382-t002:** Comparisons of surgical results.

		Group A (*n* = 86)	Group B (*n* = 59)	*p*
Duration of Surgery (min)		144.65 ± 37.4	224.65 ± 52.5	<0.0001
Conversion		1 (1.1%)	2 (3.3%)	ns
Clavien-Dindo I-II		4 (4.6%)	5 (8.4%)	ns
Clavien-Dindo III	III A	3 (3.4%)	1 (1.6%)	ns
	III B	3 (3.4%)	4 (6.7%)	ns
Mortality		/	/	ns
Readmission		2 (2.3%)	8 (13.5%)	0.01
Reoperation		4 (4.6%)	5 (8.4%)	ns
Hospital stay (days)		13.4 ± 7.3	15.3 ± 8.1	ns
Postoperative stay (days)		7.2 ± 2.3	8.8 ± 4.1	0.003

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
