# Peer review of "Sequential Approach for a Critical-View COlectomy (SACCO): A Laparoscopic Technique to Reduce Operative Time and Complications in IBD Acute Severe Colitis"

_jcm, 2020, doi:10.3390/jcm9103382_

Round 1

Reviewer 1 Report

This manuscript discusses a novel, well defined, reproducible approach for the management of severe acute colitis in inflammatory bowel disease patients.

The manuscript starts with a good introduction of subtotal colectomy indications and current operative techniques, postoperative complications and reasoning for novel new approach.

In the materials and methods section there is clear  identification of the study populations. Following the identification of the study group, the ten critical view steps are described on the  SACCo technique.

The results were discussed in a tabular format comparing group A versus group B. Although there are similarities within the groups, there was a significant difference of preoperative TNF alpha use in the groups.

In regards to the results there were significant differences in duration of surgery, readmission, and length of postoperative stay although hospital stay were similar.

This manuscript is describing a novel approach for the management of acute colitis. The critical steps are described, but would benefit from more detailed discussion and consideration of presentation in a graphic format.

The language style used in more similar to spoken english as opposed to formal medical literature style. Although the language style is understandable, it is informal in some instances.

Author Response

We thank the reviewer for his positive comments and suggestions. We hope we have “cleaned up” most of the informal English contaminations. In order to improve the take-home message of the critical-view approach, we have added two figures comprehensive of all the critical steps.

Reviewer 2 Report

I like the idea of a standardised set of steps for what can be a challenging operation.  They are nicely explained and relatively justified.  Although undoubtedly a set of standard steps for such an operation is a good idea I am not sure the paper methodology definitively proves this.  This of course relates to the retrospective nature of the study with adoption compared with a historic control.  There will be confounders, the most pertinent of which will be a learning curve.  If adoption of a lap technique occurred in 2008 and there were about 8 cases per year presumably it took a while to reach an acceptable learning curve.  This potential confounding could be reduced if the surgeons from the 2 time periods were not the same.  ie the adoption of the sequential steps benefited  less experienced surgeons (perhaps trainees) and experience was equally distributed across both time periods.  Otherwise the reduction in time and the reduction in potential adverse outcomes can be put down to more experience of the surgeons operating in the second time period.  So based on this 3 questions

1.  Where the surgeons the same in both periods?

2. Where there trainees involved in the operations and were these distributed equally?

3. Had the surgeons ( and other staff) done many cases before the first time period started ie had they reached a steady state with their learning curve

Author Response

Thank you for the positive comments and suggestions.

The IBD surgeons at “Luigi Sacco” University hospital were the same during the entire study period and performed, or assisted trainees and residents, in all the procedures from 2008. Trainees and residents participated to more than 80% of the procedure with a constant and equally distributed frequency.

Since the IBD surgeons are mainly colorectal surgeons, they came from a large experience in colorectal laparoscopic resections for cancer and diverticulitis. However, the use of laparoscopy in IBD started a bit later mainly due to a cultural reticence in operating such a complex cases by laparoscopic approach.

Substantial changes have been added to the manuscript in order to address the Reviewer requests.

Round 2

Reviewer 2 Report

Thanks for response and changes. 

Only one minor comment.  I appreciate the adoption of the SACCO technique occurred in 2015 and assume since then all patients undergoing urgent or emergency alp subtotal had this technique implemented.  Before that there was no such system.  This is indicated in the methods.  I would suggest this could be reiterated in the Results to really clarify?  ie

Among the remaining 145 patients, the SACCo technique was adopted in all 86 patients operated on after January 2015 (59.4%, Group A).  The 59 patients (40.6%) operated on before this date were treated with an unstandardised laparoscopic approach and made up the control (Group B).

Author Response

Thank you very much for your positive comment.

We modified the manuscript inserting your correction.